# Polymeric Interlayer in CdS-Free Electron-Selective Contact for Sb_2_Se_3_ Thin-Film Solar Cells

**DOI:** 10.3390/ijms24043088

**Published:** 2023-02-04

**Authors:** David Rovira, Eloi Ros, Thomas Tom, Maykel Jiménez, José Miguel Asensi, Cristobal Voz, Julian López-Vidrier, Joaquim Puigdollers, Joan Bertomeu, Edgardo Saucedo

**Affiliations:** 1Departament d’Enginyeria Electrònica, Universitat Politècnica de Catalunya (UPC), 08034 Barcelona, Spain; 2Departament de Física Aplicada, Universitat de Barcelona, 08028 Barcelona, Spain; 3Institute of Nanoscience and Nanotechnology (IN2UB), Universitat de Barcelona, 08028 Barcelona, Spain

**Keywords:** thin-film photovoltaics, conjugated polyelectrolyte, dipole, selective contacts

## Abstract

High open-circuit voltage in Sb_2_Se_3_ thin-film solar cells is a key challenge in the development of earth-abundant photovoltaic devices. CdS selective layers have been used as the standard electron contact in this technology. Long-term scalability issues due to cadmium toxicity and environmental impact are of great concern. In this study, we propose a ZnO-based buffer layer with a polymer-film-modified top interface to replace CdS in Sb_2_Se_3_ photovoltaic devices. The branched polyethylenimine layer at the ZnO and transparent electrode interface enhanced the performance of Sb_2_Se_3_ solar cells. An important increase in open-circuit voltage from 243 mV to 344 mV and a maximum efficiency of 2.4% was achieved. This study attempts to establish a relation between the use of conjugated polyelectrolyte thin films in chalcogenide photovoltaics and the resulting device improvements.

## 1. Introduction

The development of green and low-cost energy sources, such as photovoltaic (PV) technology, has appeared as an urgent need in recent years [1]. This has been encouraged by the present context of global warming caused by economic dependence on fossil fuels. Silicon-based solar cells have played a major role in this ongoing transition to carbon neutrality. This technology has dominated the solar market during the last decades [2]. This was achieved by providing highly efficient devices (reported efficiencies of over 26% [3]) with a constant reduction cost of final modules [4]. A levelized cost of energy (LCOE) of 3.3 ¢/kWh for fixed axis photovoltaic plants had been estimated [5]. However, as the silicon technology approaches its theoretical maximum efficiency [6], issues related to its carbon footprint must be addressed for it to become a truly sustainable technology. High-temperature furnaces used in wafer fabrication through the Czochralski method [7] or dopant diffusion steps in commercial technologies such as passivated rear emitter contact (PERC) are some examples. In this context, chalcogenide semiconductors have attracted attention as one of the candidates for the next generation of thin-film absorbers. This is mainly due to their low-cost, thin-film, industrially up-scalable sputtering deposition and use of earth-abundant materials. Currently, CdTe-, CIGS-, and CZTS-based thin-film solar cells (TFSCs) are reaching maturity. Power conversion efficiencies (PCEs) of 22.1%, 23.4%, and 13.0%, respectively, have been reported [8]. However, there are some drawbacks that make it difficult to commercialize this class of solar cells. The toxicity and logistics complexity of cadmium [9] or the difficulties in phase control of CIGS and CZTS absorbers are good examples. This has led to a greater effort in the investigation of new absorbers based on ternary and binary chalcogens. Antimony selenide (Sb_2_Se_3_) is a photovoltaic semiconductor which, due to its good optoelectronic properties, is becoming more prominent in TFSCs. A maximum efficiency of 10.12% for a nanostructured solar cell has been reported for this PV technology [10,11].

Sb_2_Se_3_ is a quasi-1D (Q-1D) indirect semiconductor belonging to the family of inorganic binary V_2_-VI_3_ compounds. Among other properties, it has a tunable band gap close to the theoretical optimum of 1.4 eV [12]. Furthermore, energies between 1.0 and 1.20 eV for the indirect band gap and between 1.17 and 1.30 eV for the direct can be obtained for Sb_2_Se_3_ compounds. The small difference between indirect and direct band gaps (~0.1 eV) provides a high absorption coefficient to antimony selenide. In addition, its viability for large-scale applications is supported by the compounds’ abundance in the earth’s crust (0.2 and 0.05 ppm) [13] and their non-toxicity. Moreover, this absorber presents a moderate melting point (~608 °C) and low crystallization temperature (<350 °C) [14]. Thus, thanks to the low-temperature processes involved, a reduced environmental impact is associated with its synthesis. An important feature of this material is that, for a fixed composition, a single stable phase is obtained. Therefore, problems with secondary phase formation are avoided, and a wide range of synthesis techniques are accessible [15,16,17]. Furthermore, Sb_2_Se_3_ has a preferential 1D orthorhombic crystal structure. Quasi-1D sheets of covalently bonded (Sb_4_Se_6_)_n_ nanoribbons stack together by van der Waals forces with (001) orientation [13]. This fact confers to this absorber benign grain boundaries with a low density of dangling bonds [18] and high anisotropic conductivity. The greatest charge mobility is found along the c-axis, i.e., parallel to (Sb_4_Se_6_)_n_ nanoribbons [19,20].

Currently, the most common configuration for Sb_2_Se_3_ solar cells is p-i-n geometry, i.e., hole transport layer (HTL)–absorber–electron transport layer (ETL). Typically, layers in this configuration are deposited using a bottom-up method. First, the back contact and hole transport layer (HTL) are deposited over a rigid or flexible substrate (soda–lime glass, polymers, metallic foils, etc.). On top of that, the absorber is deposited, followed by the stack consisting in the electron transport layer (ETL) and the top transparent electrode (TE). The most efficient devices obtained with this architecture employ a CdS compound for the ETL. This layer, commonly known as a buffer layer, is typically deposited by a chemical bath process. Gang Li et al. [21] reported an efficiency of 6.5% for a planar heterojunction TFSC incorporating a Cd_0.75_Zn_0.25_S buffer layer for a better energy band alignment. However, the photovoltaic conversion efficiency of these solar cells are still far from the theoretical results of 16.5% for the Sb_2_Se_3_/CdS direct junction [22] and the 32.23% of the Shockley–Queisser efficiency limit [12]. This limitation has been attributed to two main causes. On the one hand, the bulk and interface recombination losses that decrease the open-circuit voltage (V_oc_), and, on the other hand, CdS optical absorption in the blue-wavelength region that generates photocurrent losses [23]. Furthermore, it has been found that in these types of cells, the diffusion of Cd and S atoms along the space between (Sb_4_Se_6_)_n_ ribbons destabilizes the buffer/absorber junction [24]. In addition to Cd toxicity and environmental risks, a CdS replacement for an efficient electron transport layer is an interesting approach to improve the efficiency and stability of this kind of solar cell. Wider band gap materials than CdS such as ZnMgO, SnO_2_, ZnO, or TiO_2_ [25,26,27] have been proposed to increase the short-circuit current density (J_sc_). In particular, in this study we use ZnO as an ETL due to its abundance, the existence of different industrially scalable deposition techniques (spraying [27,28], atomic layer deposition [29], and solution processes [30,31]), and its non-toxicity. However, the use of ZnO as the only electron-collecting layer has failed to improve the efficiency and open-circuit voltage that is obtained with the use of CdS. Probably, an increase in surface recombination due to interfacial defects together with band misalignments are the origin of the worst performance obtained with the use of ZnO. Different approaches have been proposed to solve these problems. Strategies such as the incorporation of Mg and S in the ZnO buffer layer have been effective in the reduction of recombination losses [28,29]. Band alignment engineering is another approach to improve device performance. In this study, we explore the use of an interfacial dipole to control the energy alignment between the different layers whose function is to collect electrons while blocking the conduction of holes. In particular, we propose the use of branched polyethylenimine (b-PEI), a conjugate polyelectrolyte, as an interfacial dipole. Conjugated polyelectrolytes (CPEs) have been used in different photovoltaic technologies as an approach to avoid or lower energy barriers to charge carrier transport. In addition to the large number of CPE compounds, its compatibility with low-cost solution-processable manufacturing techniques motivates its election. Different levels of selectivity of the charge carrier can be obtained depending on the chosen CPE. Thus, in the ETL part, the use of polymers such as poly [(9,9-bis(3′-(N,N-dimethylamino)propyl)-2,7-fluorene)-alt-2,7-(9,9-dioctylfluorene) (PFN) or polyethylenimine ethoxylated (PEIE) has been reported. In contrast, poly(3,4-ethylenedioxythiophene) polystyrene sulfonate (PEDOT:PSS), 2,2′,7,7′-tetrakis[N,N-di(4-methoxyphenyl)amino]-9,9′-spirobifluorene (spiro-OMeTAD) or poly[bis(4-phenyl)(2,4,6-trimethylphenyl)amine] (PTAA) are the most studied for HTL stacks [30,31,32,33,34]. These organic interlayers have shown to decrease the contact resistivity and enhance the contact selectivity of photovoltaic devices. This results in an increased short-circuit current (J_sc_) and open-circuit voltage (V_oc_) [35,36].

This study first summarizes the deposition and characterization of b-PEI on top of polished flat crystalline silicon substrates. Afterwards, its efficiency as an electron transport layer is studied and a simple model explaining its role in the ETL interface is proposed. Next, the same approach is used to fabricate antimony selenide-based solar cells. Electrical measurements performed on diode structures allowed us to compare CdS, ZnO, and ZnO/PEI ETL contacts. The results shows that the incorporation of the polymeric interlayer results in a significant improvement in the electrical behavior of the ZnO buffer layer, reaching a performance similar to that of CdS. Finally, CdS-free solar cells, using the Sb_2_Se_3_/ZnO/b-PEI/AZO ETL configuration, were fabricated and characterized.

## 2. Results and Discussion

### 2.1. PEI Characterization

#### 2.1.1. Optical Characterization

X-ray photoelectron spectroscopy (XPS) and energy-dispersive X-ray spectroscopy (EDS) were performed on PEI films deposited over a chemically polished silicon substrate to ensure homogeneous coverage. From the XPS results (Figure 1), one can distinguish chemical bonding states different from the b-PEI chains. The presence of carbon–oxygen bonds can be seen in the C1s spectrum. In the nitrogen spectrum, a mix of sp^2^/sp^3^ hybridization is observed. In polyelectrolytes such as polyaniline and polypyrrole, the C-O signal is sometimes associated with surface oxidation of some sort [37,38]. However, local composition analysis by means of high-resolution transmission electron microscopy (HRTEM) combined with electron energy loss spectroscopy (EELS) of the b-PEI films suggests that the nitrogen and oxygen specimens could be from a bilayer structure [39]. This picture is consistent with the interaction of the cationic polymer (PEI^+^) via the sp^3^-hybridized nitrogen specimens and their respective counterions from the solvent (EtOx^−^). The solvent ethanol would behave as a Brønstred acid (proton donor) by interaction with the b-PEI, a reported Lewis base (electron donor). This results in an ethanolate (that can potentially become its respective alkoxide) and a protonated amino group. The negatively charged nature of the reduced solvent molecule will be electrostatically drawn to the positively charged amino group. A molecular dipole between them is generated. Thus, counterions from the solvent are present in the polymeric film after the solvent’s evaporation. This picture is consistent with the presence of C-O in the C1s spectrum and the sp^3^ hybridization in the nitrogen spectrum.

EDS data (Appendix A) indicate the presence of an organic layer of ~1–2 nm thickness between the silicon substrate and the aluminum electrode. From these results, one can observe the coexisting presence of oxygen and carbon signals in the organic layer. The resulting experiment indicates that oxygen is well integrated within the organic film. Nevertheless, not enough resolution is obtained to observe the formation of a charge bilayer. The presence of a nitrogen signal in the XPS spectra should also point out that the observed carbon and oxygen signals in the EDS are indeed evidence of the branched polyethylenimine and not a surface oxide.

The use of these ultrathin films avoids undesired optical absorption that could reduce the generated photocurrent. In addition, optical transmittance studies can be performed to determine the optical band gap of b-PEI films through a Tauc plot (Figure 2). A wide band gap of 5.337 eV is obtained using Tauc’s relation for direct band gap [${\left(\alpha h\upsilon \right)}^{2}=C\times \left(h\upsilon -E_{g}\right)$] [40], where hυ is the incident photon energy, *C* is the proportionality constant, and $E_{g}$ is the band gap energy. This high value corresponds to an expected insulating behavior and optimum transparency in the visible spectrum of b-PEI.

#### 2.1.2. Electrical Characterization

Dipole strength tuning has proven to be relatively easy. In this case, the polymer was deposited using different solvents over n-type silicon. The resulting contact resistivity obtained through TLM is plotted in Figure 3a. The exact values are shown in Appendix A. Four samples of c-Si/b-PEI/Al with toluene, ethanol, ethanol/water mix, and methanol as solvents were fabricated. The TLM curves for each sample are shown in Appendix A. As can be seen, a decrease in the resistivity is found when the molecular dipole moment of the solvent [41] increases. This dependence could be related to polymer–solvent interaction. Higher polar solvents could behave as better proton donors, increasing the ratio of counterions and protonated amino groups in the film. Thus, an increase in the charge density within the film is obtained.

The minimum contact resistivity is obtained with methanol as the b-PEI solvent. However, in the following device fabrication, ethanol is preferred. Since the bulk resistance for the Sb_2_Se_3_ dominates the total device resistance, the difference in contact resistivity from $10^{-2}\;Ω\cdot {\mathrm{cm}}^{2}$ to $5\cdot 10^{-2\;}Ω\cdot {\mathrm{cm}}^{2}$ has a low effect on cell performance. Therefore, ethanol is considered primarily for safety reasons. Furthermore, thanks to the extensive study of b-PEI in ethanol solution, its use facilitates the comparison of results.

Similarly, through UPS one can observe how the substrate’s work function (WF) reduces when the b-PEI layer is deposited. This reduction can be further accentuated with the incorporation of a semitransparent aluminum capping (Figure 4). The initially found work function of 4.11 eV for c-Si is reduced to 3.28 eV when the b-PEI film is incorporated and to 3.08 eV after the 10 nm thick semitransparent Al layer is deposited on top of the b-PEI. The obtained UPS spectra is consistent with the reported increase in passivation of Si/b-PEI after Al metallization [39]. This would suggest that aluminum presence promotes the orientation of the dipoles. This reorientation phenomenon is also proposed in other studies. Yun Juwon et al. [30] proposed a dipolar interaction, governed by dielectric constants, directly between the PEIE molecules and the contact layers. Aluminum deposition over the b-PEI could behave in a similar manner. A preferential orientation of the dipoles is promoted, enhancing the electrostatic potential. Furthermore, this stronger dipolar interaction will increase the transfer of electrons from the cathode to the semiconductor interface [43]. Dipole reorientation and semiconductor electron doping are the main candidates to explain the results obtained from UPS. Following this hypothesis, surface traps get neutralized from the charge transfer and the semiconductor’s band bends. A reduction in the Fermi-level pinning (FLP) and an improvement in contact selectivity are expected (Figure 3b). For further information, see references [42,44,45]. Solvents with higher molecular dipole moments present a higher negative charge density in their oxygen functional groups. This will increase the dipole field between the amino groups of the b-PEI and the counterions. Thus, a higher charge transfer to the semiconductor and improved resistivity and selectivity are obtained. This mechanism is similar to the use of low-work-function metals such as calcium or magnesium to achieve ohmic electron contacts [46,47].

### 2.2. Sb_2_Se_3_ Diode Devices

The transfer of the b-PEI interlayers as the ETL in antimony selenide devices is explored in this section. Initially, an ETL design using the stack b-PEI/Al was evaluated. A 300 nm aluminum film was used as the electrode for all diode devices. With that, the authors tried to show the applicability in Sb_2_Se_3_ of the dipolar model for b-PEI interlayers. Four different solutions of b-PEI (0.001%, 0.01%, 0.1%, 1%) in ethanol solutions were coated over the Sb_2_Se_3_ samples. Slightly selenized molybdenum was used as the rear hole transport layer (HTL). The J-V curves can be found in the Appendix A. The curves are fitted to a single-diode model with series and shunt resistance. With that, one can split the dependence of the contact performance with b-PEI concentration (i.e., film thickness). Figure 5 shows the series resistance and the diode ideality factor as a function of b-PEI concentration. A parabolic (i.e., “V”-shaped) profile with a minimum in the 0.01 wt.% sample is observed. This type of behavior indicates the combination of two transport mechanisms. These could potentially be thermionic emission and direct tunneling. The first would be responsible for the resistivity increase in concentrations below the optimum. When the thickness of the polymer layer is excessively thin, the dipole effect is not capable of preventing the surface pinning of the metal contact. Thus, an energetic barrier grows with the decrease in the polymer’s thickness. Similarly, when the organic layer is far too thick, the contact resistivity increases. In this case, the resistance can be introduced by the conduction through the thicker b-PEI film. As the resistivity increases after passing the 1–2 nm thickness mark (i.e., 0.01–0.1%), direct tunneling makes sense as one of the predominant conduction mechanisms, limiting the transport and increasing the resistivity in this structure.

As shown in Figure 5 and Appendix A, an optimum concentration of 0.01% is found for the b-PEI layer in this b-PEI/Al contact. The high ideality factor can be associated with the high recombination at the interface and probably a lower contact selectivity. Low asymmetry in the J-V curves from direct to reverse bias is observed in Appendix A. This could indicate a deficient selectivity for the minority carriers in the ETL, attributable to high pinhole density.

In order to enhance the diffusion of minority carriers, the dipole was combined with a buffer layer. A reduction in the pinhole density is expected, helping to improve the shunt resistance. CdS, the most widely used electron-selective layer in chalcogenide technology, has been compared to ALD-deposited ZnO films. Thanks to ZnO’s wider band gap, a higher energetic barrier for holes is expected compared with CdS. Thus, a better selectivity should be observed. Further, the possibility to deposit it through ALD, as well as the transparent electrode (AZO), facilitates its large-scale implementation. In addition, ZnO has highly tunable electrical properties for oxygen vacancies, extending its versatility. Figure 6 shows the J-V curves of the fabricated devices incorporating buffer layers. A total of four samples are represented, two of them using CdS and the rest ZnO. In addition, the PEI film is incorporated into one sample from each pair. The fitting parameters of these curves to a single-diode model with series and shunt resistance are shown in Table 1.

In accordance with the model given in Section 2.1.2, the organic layer should be below the electrode. Therefore, the passivation effect of the dipolar film can be enhanced with the carrier transfer from the electrode. This is a different approach from other studies where the organic film is in between the absorber and the buffer layer [49]. Due to the incorporation of this film, the holes encounter a higher energy barrier, as shown in Figure 7. This will result in an enhancement in the diode’s ideality factor. A shift to lower b-PEI concentrations is observed in the minimum values for series resistance and ideality factor (Appendix A) when compared with diodes without buffer layers. The deposition of b-PEI on top of a selective contact such as CdS or ZnO instead of antimony selenide results in a different optimum value. In this case, a lower concentration of 0.001% is found.

The presence of a conjugated polyelectrolyte produces an improvement in most of the diode parameters. It is especially significant in the series resistance and the ideality factor, and it appears to be independent of the buffer layer. This is consistent with the reported decrease in the specific contact resistance and the increase in passivation [36,39,50]. The incorporation of b-PEI depletes the band of the interlayer. Tunneling through the conduction band for electrons is facilitated, while the barrier for holes is increased (Figure 7).

The same diode ideality factor is obtained for the ZnO/b-PEI/Al and the CdS/Al contacts. However, the first shows an important improvement in the series resistance. Inhomogeneities in the absorber samples can be responsible for shunt differences. Its strong sensitivity to fabrication conditions makes the appearance of pinholes a recurrent issue [12,51,52]. In light of these findings, ZnO/b-PEI/Al is identified as a prospective CdS-free, non-toxic, and earth-abundant structure capable of competing with the conventional CdS buffer layer.

### 2.3. Sb_2_Se_3_ Photovoltaic Devices

Finally, the bilayer cathode ZnO/b-PEI was implemented in a series of photovoltaic structures. As the transparent electrode, zinc oxide doped with aluminum (AZO), was chosen. This transparent conducting oxide (TCO) can be deposited by ALD. This is a soft deposition technique that allows conformal deposition and up-scalability. Furthermore, the presence of aluminum in the TCO supports the choice since Al might have a prominent role in dipole orientation. Figure 8 shows the schematic film representation of the device as well as the band structure of the photovoltaic device. The wide band gap of both intrinsic and doped zinc oxide layers leads to a high energetic barrier for holes. In addition, the voltage provided by the dipole film prevents pinning at the surface of the ZnO. This facilitates the tunneling of electrons through its conduction band into the TCO. It has been noticed that thicker layers of b-PEI are required compared with diode devices. This is due to the lower carrier density of AZO compared with aluminum [53]. Thus, more dipoles will be needed to affect the band structure and provide similar passivation and selectivity.

Consequently, a reference sample without organic film and three devices with different concentrations of b-PEI solutions in ethanol (0.01 wt.%, 0.05 wt.%, and 0.1 wt.%) were fabricated. Figure 9 shows the J-V curves of the fabricated devices under 1.5 AM illumination and their corresponding power density characteristics. The obtained photovoltaic parameters are summarized in Table 2.

It seems that open-circuit voltage (V_oc_) improves as the b-PEI layer thickness increases. This would point to an enhanced surface passivation and the successful elimination of interface energy barriers from FLP. The presence of the organic interlayer translates into a 57% enhancement in power conversion efficiency. A PCE of 2.411% for the sample with the largest polymer concentration was obtained. In addition, the presence of the b-PEI interlayer increases the FF and J_sc_, with maximum values of 35.36% and 22.93 mA/cm^2^ in the 0.05 wt.% sample, respectively. The relationship between FF, J_sc_, and b-PEI concentration appears to peak between 0.05 and 0.1 wt.%. Nevertheless, an enhancement in AZO conductivity could potentially move these values as they are related to the strength of the dipole effect.

The optimal concentration in the solar cells is pushed to higher concentrations relative to the experiments with the aluminum contact. Higher concentrations of b-PEI are necessary as the equilibrium state between the semiconductor and the electrode changes when replacing aluminum with AZO. However, this shift to higher concentrations is limited by a trade-off between the rise in V_oc_ and the decrease in J_sc_. Thicker layers will increase the dipolar voltage. At the same time, the series resistance of the device will increase due to the dielectric behavior of b-PEI.

Nonetheless, the causes for the FF to remain under 40% should be further investigated to understand whether it is a problem of the absorber or the contacts. The most probable cause for the decreased FF values could be the shunt leaks coming from pinholes in the absorber. This could potentially be improved by the deposition of thicker or less conductive ZnO films.

Finally, the effect of the b-PEI and ZnO layers in the absorbance spectrum can be studied from the quantum efficiency (Figure 10). First, the ZnO layer plays a crucial role in the antireflective quality of the window stack. Samples containing it exhibited a flatter spectrum and absorbed the visible range to a higher extent. EQEs remain above 60% from wavelengths smaller than 700 nm. In contrast, the sample without the ZnO layer experiences a drop in EQE at approximately 550 nm. Then, the effect of b-PEI is mostly seen on lower wavelengths by a slight shift in the absorbance maximum. Thus, a greater photogenerated current will be obtained for the ZnO/b-PEI sample, as we have already seen in the J-V curves. From the transfer matrix method (TMM) analysis, we determined that the insertion of a PEI interlayer between the AZO/ZnO should increase the reflectivity of the stack. This indicates that the increase in photogenerated current for thin b-PEI films (lower than 0.1 wt.%) has an electrical rather than an optical origin. The increase in selectivity introduced by the polymeric layer increases the ratio of electrons extracted per incident photon, even when the number of photons reaching the absorber is slightly lower due to the higher reflectivity.

Moreover, with these EQEs one can extract the band gap of the antimony selenide. A value of E_g_ = 1.305 ± 0.012 eV is obtained, consistent with the literature [54,55].

## 3. Materials and Methods

### 3.1. Preparation of Sb_2_Se_3_ Thin Films

The first step in the synthesis of the absorber material consisted in depositing Mo on soda–lime glass (SLG/Mo) by magnetron sputtering. After that, about 280 nm of Sb (Sb shots, Alfa Aesar, 1–3 mm) was evaporated on the deposited Mo. This thermal evaporation was carried out in an Oerlikon Univex 250 in a vacuum of 10^−5^ mbar with a rate of 1 nm/s. This was followed by a reactive annealing in a furnace under a selenium atmosphere. The samples were then set in 23 cm^2^ graphite boxes containing two crucibles with 25 mg of selenium (12.5 mg each, Alfa Aesar, Se powder 200 mesh). The temperature was increased with a ramp of 20°/min up to 320 °C, reaching a final pressure of 500 mbar inside the tube. The temperature of 320 °C was maintained for 30 min, followed by cooling at room temperature. The final composition of the layer depends to a large extent on the number of processes carried out on the optimization boxes. The graphite boxes were always precleaned at 650 °C for two hours. After that, optimization processes on the boxes were carried out to try to be as reproducible as possible. The chemical composition of the Sb_2_Se_3_ thin films was analyzed through XRF spectroscopy measurement. Material structure was characterized by XRD diffraction and SEM micrograph (see Appendix A).

### 3.2. Fabrication and Characterization of b-PEI Layers

Branched polyethylenimine (b-PEI) solutions 50 wt.% in H_2_O, average Mw = 750,000 by LS, were used as electron transport layer (ETL). Polyethylenimine solutions of different weight percentages, from 1 wt.% to 0.001 wt.%, were prepared using ethanol as solvent and stored at a constant temperature of 3 °C. All the chemicals were purchased from Sigma-Aldrich (Seoul, Republic of Korea). One side chemically polished (FZ) n-type c-silicon (100) wafers with a thickness of 280 μm and resistivity of 2 Ω·cm were used as substrate in the characterization experiments of b-PEI films. All wafers were treated with a diluted hydrofluoric acid 1 (*v*/*v*)% immersion for 30 s to remove the native oxide from the surface. Spin coating of b-PEI solutions was performed at 5000 rpm for 30 s. Before spin coating, all solutions were rested at room temperature for 30 min. Then, the films were annealed at a temperature of 80 °C for 2 min on a hot plate in ambient air. A 300 nm aluminum film was deposited by thermal evaporation using a shadow mask for the transmission line method (TLM) experiment [56,57]. In order to change the organic layer thickness, solutions with different concentrations of b-PEI were used while the angular speed of the spin coating remained constant. The thickness of b-PEI films of various concentrations was measured by ellipsometry. The equivalence between organic film thickness and b-PEI concentration is shown in Appendix A. Amorphous hydrogenated silicon was deposited using an elettrorava PECVD system with a combination of 36 sccms of silane and 4 sccms of methane for 10 s. The low percentage of methane prevents the formation of silicon carbide. Instead, a carbon-doped amorphous silicon film is deposited, conferring high resilience to surface oxidation.

UV–Visible–NIR spectrophotometer Lambda 950 (Perkin Elmer, Shelton, CT, USA) was employed to study the transmittance of b-PEI films spin-coated on transparent sapphire substrates. X-ray and UV photoelectron spectroscopy (XPS and UPS, respectively) measurements of the films over silicon wafers were performed with a Phoibos 150 analyzer (SPECS GmbH, Berlin, Germany) in ultrahigh vacuum conditions (base pressure of 5 × 10^−10^ mbar). XPS measurements were carried out with a monochromatic Al K-alpha X-ray source (1486.74 eV). A monochromatic He I UV source (21.2 eV) was used for UPS measurements. The energy resolution, as the FWHM of the Ag 3d_5/2_ peak for a sputtered silver foil, was 0.62 eV for XPS and 0.11 eV for UPS. The XPS spectra were analyzed using the CASA XPS software and data were fitted for C, O, N. The work function of the films was extracted from the analysis of UPS spectra.

### 3.3. Fabrication and Characterization of Sb_2_Se_3_ Devices

Zinc oxide (ZnO) and cadmium sulphide (CdS) were used as buffer layers in Sb_2_Se_3_ devices. ZnO films were deposited via atomic layer deposition (ALD) onto Sb_2_Se_3_ absorbers using an Ultratech Savannah ALD System. The films were grown with the use of diethyl zinc (DEZ, UP chemical Co., Ltd., Pyeongtaek, Republic of Korea) and deionized water vapors as precursors of zinc and oxygen, respectively. The deposition temperature was set at 130 °C, and a working pressure of 0.38 mbar was employed with a N_2_ carrier flow of 20 sccm. The zinc and oxygen precursors were contained in cylinders held at 75 °C and room temperature, respectively. One ZnO ALD cycle consisted of a water vapor pulsing (0.02 s), N_2_ purge (5 s), DEZ pulsing (0.015 s), and N_2_ purge (5 s). This cycle was repeated 300 times, resulting in film thicknesses ranging from 50 to 54 nm. A 35 kΩ/sq sheet resistance for the films was obtained with a 4-point probe. For devices with CdS buffer layer, a 50 nm thick CdS film was deposited onto the absorber by chemical bath deposition (CBD). Next, spin coating of b-PEI layers in the same conditions as in Section 2.2. was performed. However, the annealing time was increased to 6 min due to the higher thickness of the Sb_2_Se_3_ glass substrate.

After b-PEI incorporation, and for diode devices, aluminum electrodes of 300 nm thickness and 0.1 cm^2^ area were deposited by thermal evaporation. In contrast, for photovoltaic devices, the deposition of the transparent conductive window layer (TCO) via ALD (Ultratech Savannah) was followed. Zinc oxide doped with aluminum (AZO) was grown with the use of DEZ, deionized water vapors, and trimethylaluminum (TMA, UP chemical Co., Ltd., Pyeongtaek, Republic of Korea) as precursors of zinc, oxygen, and aluminum, respectively. The deposition temperature was set at 150 °C, and a working pressure of 0.4 mbar was employed with a N_2_ carrier flow of 20 sccm. The zinc precursor was contained in a cylinder held at 75 °C, whereas the aluminum and oxygen ones were at room temperature. One alumina (Al_2_O_3_) ALD cycle consisted of TMA pulsing (0.05 s), N_2_ purge (5 s), water vapor pulsing (0.02 s), and N_2_ purge (5 s). The same zinc oxide (ZnO) ALD cycle of the buffer layer was used here. The recipe followed for AZO films repeats 45 times the sequence of 19 ZnO ALD cycles by one Al_2_O_3_ ALD cycle, i.e., ((19 + 1)45). With this deposition, we obtained an AZO film of thickness and sheet resistance ranging from 138 to 146 nm and 184 to 230 Ω/sq, respectively. Thereafter, lateral isolation was performed by means of microdiamond scriber MR200 OEG to have cells of 0.1 cm^2^. Finally, small silver dots at each device were thermally evaporated in order to facilitate the contact during the characterization processes.

The electrical characterization of the samples was performed several times to assess any possible light soaking effect. The current density–voltage (J-V) characteristics of the cells were measured under standard conditions (100 mW/cm^2^, AM 1.5G spectrum) using 94041A solar simulator (Newport, Irvine, CA, USA). Finally, external quantum efficiency (EQE) analysis was performed with a QEX10 setup (PV Measurements, Point Roberts, WA, USA).

## 4. Conclusions

In this study, we have introduced branched polyethylenimine (b-PEI) as a polymeric electron transport layer in Sb_2_Se_3_ planar heterojunction TFSC. Optical and electrical characterizations of b-PEI thin layers were performed. The results obtained suggest that the working principle of this organic ETL is the arrangement of interfacial molecular dipoles between amino groups in the polymeric chain and counterions coming from the solvent. Thus, the contact performance has the potential to be upgraded via chemical tuning of the b-PEI solution. The first attempts of implementing b-PEI thin films directly between the chalcogenide absorber and a metallic electrode in the metal/dipole/semiconductor structure showed an enhanced dipole behavior and reduced series resistance. Nevertheless, the low ideality factor indicates high surface recombination of minority carriers when only b-PEI is used. To further enhance diode performance, CdS and ZnO films were added to the structure in combination with b-PEI interlayers. The contact’s resistivity and selectivity were improved, finding its optimum with a b-PEI solution of 0.001 wt.% in diode devices. The series resistance dependence on thickness was also studied. From this, a competition of two mechanisms affecting the junction was discerned. From these devices, it was seen that a bilayer of ZnO/b-PEI could be a viable alternative to CdS electron-selective contacts.

Finally, a CdS-free Sb_2_Se_3_ photovoltaic test structure incorporating a ZnO/b-PEI/AZO stack was proposed. A large improvement in the open-circuit voltage of these solar cells was found of over 40% with respect to reference samples without the organic interlayer. This finding completely agrees with the effect of conjugated polyelectrolytes reported in the literature. Additionally, it opens the possibility of improving CdS-free selective contacts by the combination of heterojunction and dipole thin films. A promising performance with a PCE of 2.41% was obtained which could be further enhanced both by contact and absorber optimization.

## Figures and Tables

**Figure 1 ijms-24-03088-f001:**
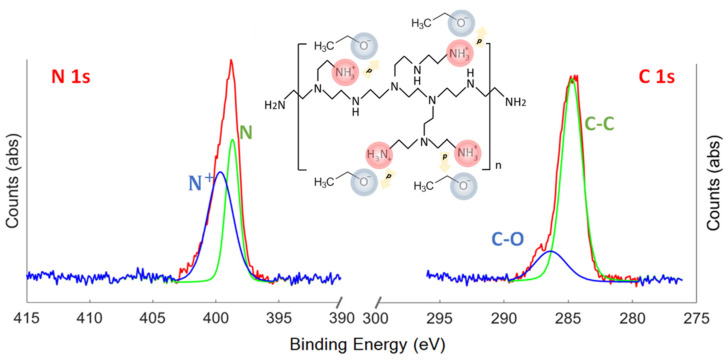
XPS spectrum of the b-PEI interlayer after 0.01 wt.% solution in ethanol deposition over c-Si substrate and schematic representation of XPS results.

**Figure 2 ijms-24-03088-f002:**
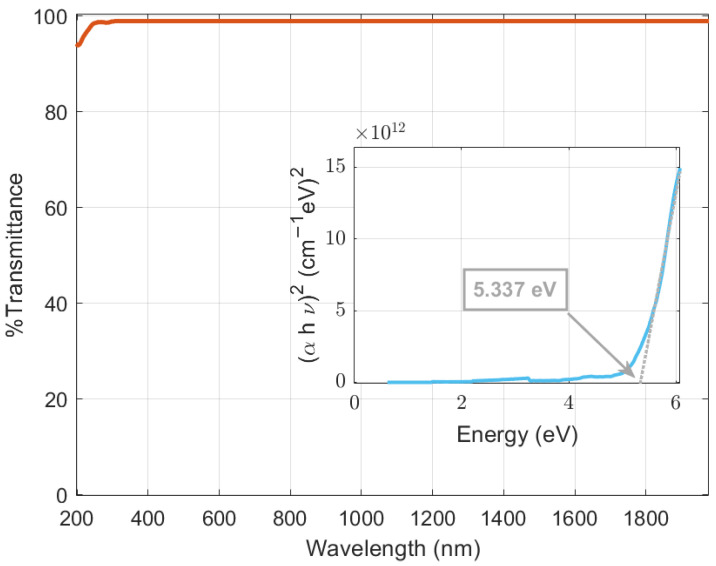
Tauc plot of the photospectroscopy transmittance of a 1 nm b-PEI film on sapphire.

**Figure 3 ijms-24-03088-f003:**
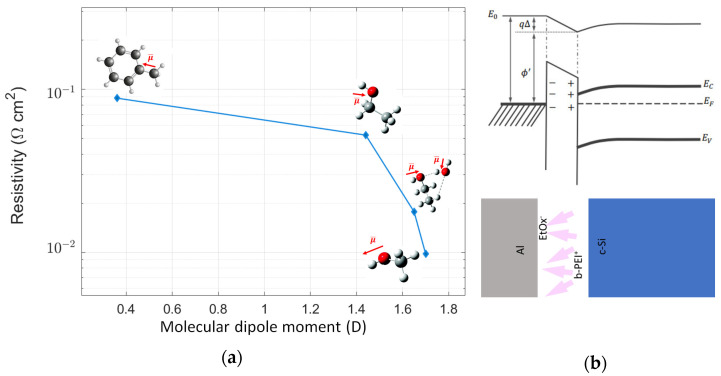
(**a**) Contact resistivity as a function of the solvent’s molecular dipole moment for 1 nm thick b-PEI films, from lower to higher molecular dipole moment, with solutions of b-PEI 0.01 wt.% in toluene, ethanol, ethanol/water mix, and methanol. (**b**) Band bending effect of b-PEI interlayer in ethanol [42].

**Figure 4 ijms-24-03088-f004:**
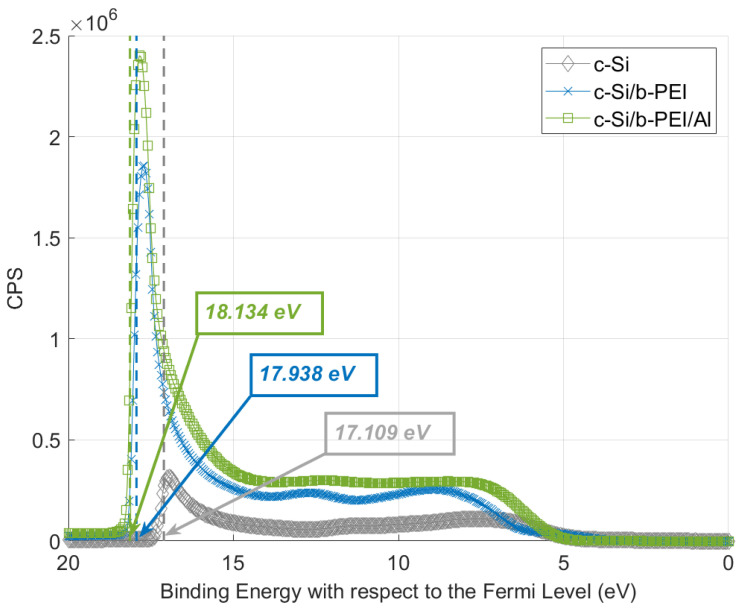
UPS spectrum for c-Si, c-Si/b-PEI (1 nm), and c-Si/b-PEI (1 nm)/Al (10 nm) and its corresponding secondary electron cutoffs (SEC). From this, one can extract the work function (WF = ħ*ω* − SEC − E_f_ eV, with ħ*ω* = 21.22 eV and E_f_ = 0 eV) [48].

**Figure 5 ijms-24-03088-f005:**
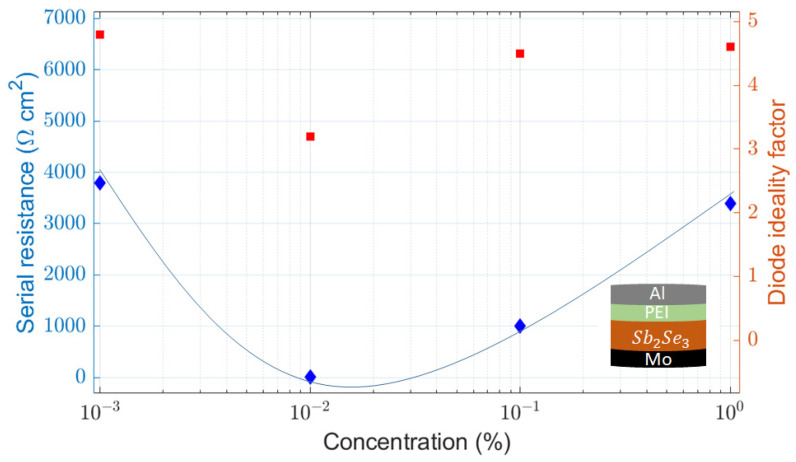
Fitted values of series resistance (blue) and ideality factor (red) for Mo/Sb_2_Se_3_/b-PEI/Al diodes with different b-PEI concentrations.

**Figure 6 ijms-24-03088-f006:**
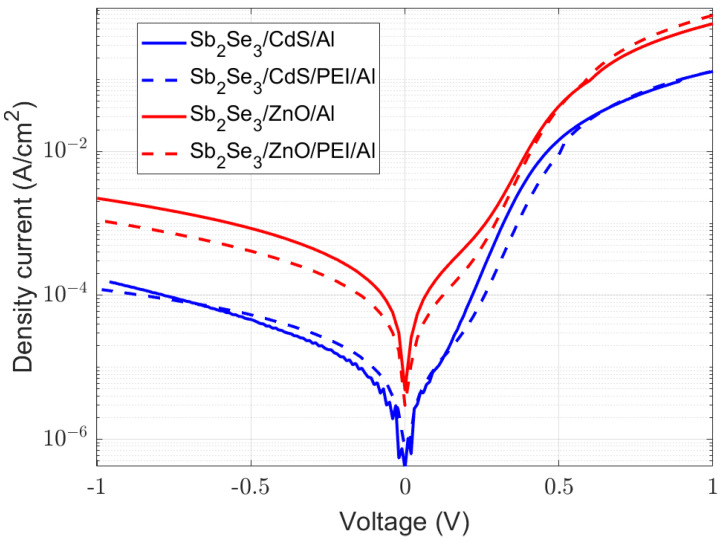
J-V curves of Sb_2_Se_3_ diodes with CdS and ZnO buffer layers with and without 0.001 wt.% b-PEI interlayer (0.5 nm thickness).

**Figure 7 ijms-24-03088-f007:**
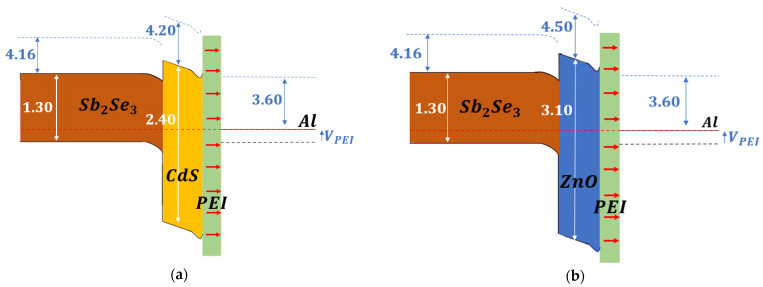
Band alignment of the diode’s ETL stacks with b-PEI film and (**a**) CdS or (**b**) ZnO as buffer layer. Energies in eV.

**Figure 8 ijms-24-03088-f008:**
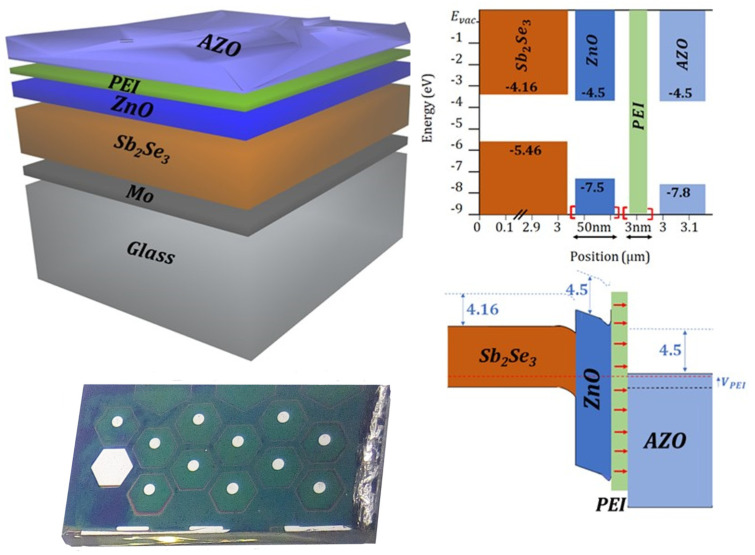
Sample and schematic design of the CdS-free cell structure with its corresponding band alignment.

**Figure 9 ijms-24-03088-f009:**
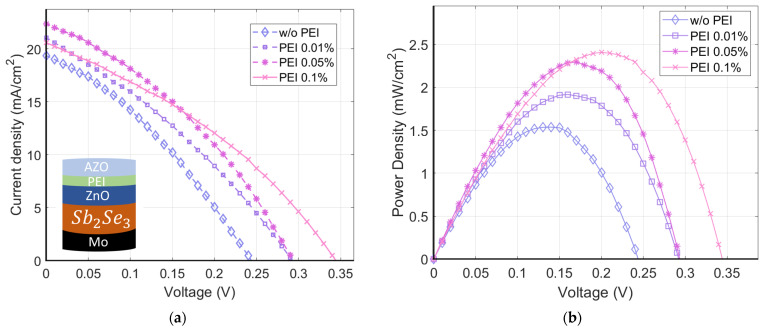
(**a**) J-V first quadrant curves under 1.5 AM illumination for Mo/Sb_2_Se_3_/ZnO/b-PEI/AZO solar cells. (**b**) Power density characteristics of the fabricated devices.

**Figure 10 ijms-24-03088-f010:**
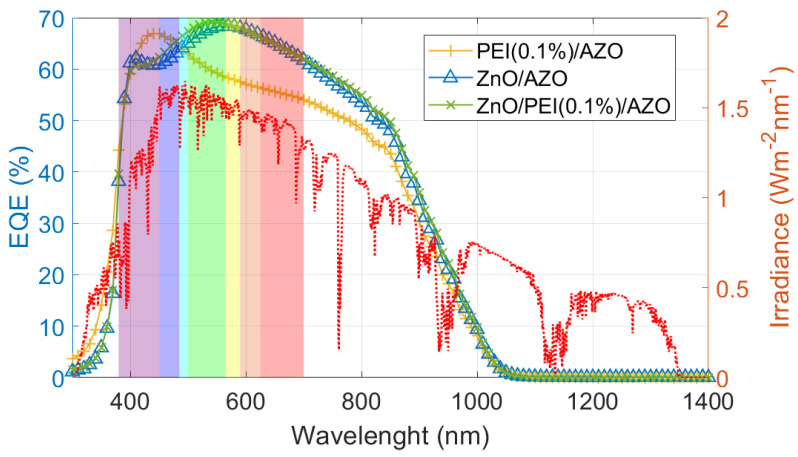
External quantum efficiency spectra of different ETL stacks in Sb_2_Se_3_ solar cells and AM 1.5 solar irradiance (red). Visible light spectrum highlighted by background colors.

**Table 1 ijms-24-03088-t001:** Fitted parameters of different Sb_2_Se_3_ diode architectures.

ETL Stack	Ideality Factor, n	Series Resistance, R_s_ (Ω·cm^2^)	Shunt Resistance, R_sh_ (kΩ)
CdS/Al	1.65	4.14	160
CdS/b-PEI/Al	1.46	3.40	115
ZnO/Al	1.70	0.65	6
ZnO/b-PEI/Al	1.65	0.50	12

**Table 2 ijms-24-03088-t002:** Photovoltaic parameters of Mo/Sb_2_Se_3_/ZnO/b-PEI/AZO solar cells for different b-PEI concentrations.

	w/o PEI	PEI 0.01 wt.%	PEI 0.05 wt.%	PEI 0.1 wt.%
Voc (mV)	243.84	292.23	293.42	344.23
J_SC_ (mA/cm^2^)	19.35	21.05	22.93	20.67
FF (%)	33.07	31.39	35.36	34.42
Efficiency (%)	1.536	1.917	2.294	2.411

## Data Availability

The data presented in this study are available on request from the corresponding author.

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
