# Peer review of "Polymeric Interlayer in CdS-Free Electron-Selective Contact for Sb_2_Se_3_ Thin-Film Solar Cells"

_ijms, 2023, doi:10.3390/ijms24043088_

Round 1
Reviewer 1 Report
This manuscript presents a study on polymeric interlayer for antimony triselenide for solar cell applications. In this study, the authors have included key components such as fabrication of the thin film solar cells, materials characterization such as XRD, SEM, and optical characterization of the device. However, the manuscript lacks clarity and a cogent explanation of inferences and conclusions drawn from the study. This reviewer strongly recommends this manuscript be reviewed and edited by the English language editing service associated with this journal. Often the sentences in this manuscript are long and it is hard to understand what the authors want to convey. Some of the abbreviations that are used in the content do not have full form explained before or after.
1. Abstract - "Throughout characterization of the polymeric thin-film and device comparison 17
with standard CdS contacts, the work in this paper stablishes a principle behind the improvements 18
provided by the polymer and a route to further improve the results obtained." This sentence is long, and it is hard to understand the author's point of view. Please reword and replace it with short and simple sentences. what is AZO here? the complete form is not introduced anywhere before it was used in the Abstract. Please provide the complete forms of the abbreviation before using it in the textual content.
2. Introduction - line 1 to line 40- Filled with long incorrigible sentences. Please rewrite all the sentences in lines 1-40.
3. Figure 2 - Please move this figure to the supporting information section or remove it from the manuscript. This figure and the related paragraph are not particularly insightful or contribute to the body of research.
4. Figures 3 & 4 can be grouped as they are optical measurements.
5. Sb2Se3, ZnO, and CdS-based thin films are well-documented to have solar cell applications and many optical devices are ubiquitously reported. Please substantiate the novelty and originality of this study.
Reviewer 2 Report
Dear Authors, I’ve read the manuscript and I have several comments to do.
Let me say that the idea of removing CdS on this kind of cell is welcome and similar works should be worth to be read and published, however in the present form, this paper presents several weak points. Let me say that actually, the part involving the ZnO is smaller than that dedicated to the polymer, and this is in accordance with the title. However, reading the abstract one could expect more emphasis on the ZnO part of the work.
The first regards the readability of the paper.
I don’t know well if it is a requirement of the journal to place the experiments details at the end of the paper, but I think that when presenting the results, just after the introduction, the reader should know what the discussed samples are, and how they are prepared, or at least which kind of structure is investigated.
Thus if the materials and methods are required to be in section 3, then in section 2 when presenting results, the investigated structure should be recalled, at least as an image, involving at least the layers and the thicknesses, remanding to section 3 for further details. As an example, when discussing Figure 3, the reader should know the sample is PEI on sapphire, when looking at figure 4 it is necessary to know the three structures, and especially the layer thicknesses, since, as I will discuss further, this is an important parameter to understand the figure. Similar considerations can be done for each figure and text part of the results section. However, I would always prefer to read experiments before, if it is not a mandatory scheme from the journal.
The second concern is about the specific information. For example, Figure 5 describes the contact resistance of different structures produced with different solvents with different molecular dipole moments. It is not immediate to individuate which solvent produces the resistivity, thus I suggest better specifying who is who, or producing a table or a list where each solvent is associated at the dipole. Furthermore, the thickness of each layer for each dipole should be indicated, since this seems to be a critical aspect of the whole discussion. Further, I would advise placing in the Supplementary information all the TLM graphs and adding a reference to the TLM technique in the paper.
Other minor issues I would like to mention:
-the coherence in the terms: for example, on page 2 line 46 it is mentioned Antimony selenide, and on page 3 line 109 it is mentioned antimuonium selenide. On page 6 line 163 it is mentioned contact resistance for figure 5, but on the figure, it is referred to as resistivity. They are two different quantities, as the first refers to the effective resistance measured in the TLM technique, while resistivity is per unit area. Please check the whole paper for differences in terms and adopt the same in the paper.
-Page 3 line 116 and further: please comment about the use of a silicon wafer for XPS and UPS measurements instead of another conductive substrate (eg: something covered by gold), as this could make more sure the work function of the substrate and would avoid any doubt about the eventual oxidation of the silicon, as discussed at page 4 lines 138-145.
-On page 14 line 352 employing methane and silane in PECVD one does not obtain amorphous silicon but a silicon carbide. Nevertheless, it is not clear where this layer is deposited, in which structure, and the purpose.
-On page 14 section 3.3 it is not clear at the end the aspect of the cell. Where is the AL dot located? How much is the active area?
-page 8 line 198. If from figure5 it is clear the lower contact resistance is for PEI deposited with methanol, why for the cells it is used ethanol?
Finally, let me express my stronger perplexities.
1)UPS measurements and WF values.
In the text on page 7 line 175, the WF for silicon is mentioned to be 4.7eV. However, on page 13 line 343 it is specified that silicon is n-type 2 ohm cm. This makes it evident that its WF can’t be 4.7 eV (which is typical of intrinsic silicon) but something around 4.1-4.15eV. Moreover, considering that the electron affinity of silicon is 4.05 eV, it appears impossible to mention WF like 3.31 or 2.49 eV. This arises I think from an incorrect interpretation of the UPS measurements. Usually, such spectra are expressed as a function of kinetic energy, and already rescaled by the bias. Nevertheless, even in binding energy, the problem is in the proposed formula for the WF calculation. The quantity W is erroneously considered, because also in the case of a semiconductor it has to be considered from the Fermi level, at 21.22 eV.
The WF is in this way the intercept value with the x of the electron secondary cutoff.
In this way, looking at the graph for c-si in figure 4, considering -10 as the upper point of 21.22 eV value, the secondary electrons cutoff is around 7.2 eV, thus WF= 21.22-10-7.1= 4.12eV that is a number which makes sense. More details can be found for example in the following review: J.W Kim and A. Kim, Current Applied Physics 31 (2021) 52-59 and references therein. I am aware that in literature many works report calculations similar to yours, which however are wrong.
Further, it is necessary to understand the thicknesses of PEI and Al deposited on the structure since the probe of UPS is able to penetrate a few (2-3) nm inside the sample, that is for the sample with c-Si/PEI, being PEI layer of 1-2 nm, then it can be supposed the analysis regarding also the silicon.
However, in the case of the structure completed with the Al layer, this assumption starts to fall, for two reasons. The first is that on page14 lines 348 or 376 Al layers are mentioned to be 300 nm. In this case, only the Al would be analyzed. The second is that even if the metal layer was 1 nm, the WF of the material with lower WF would dominate the cut-off value.
Thus I think all this section and the relative deduction on page7 lines 172-193 should be revised. In particular, the role of the charges should be reconsidered.
In n-type silicon the majority carriers are electrons. A reduction in WF should mean a shift of the Fermi level to the conduction band, that is an increased energy of the electrons, which happens, let’s say because there are more electrons, which then accumulate at the interface. On page 7 lines 183-193 an orientation of dipole is called to justify this behaviour together with an electron transfer from the cathode (of the UPS equipment, I suppose) to the silicon. And at line 190 this charge is referred to as a negative charge. However, to promote an electric field-induced electron accumulation to the surface in the c-Si, the charge at the interface must be positive. Thus I think a better explanation of this phenomenon (if confirmed by correct WF calculations) should be provided, also specifying why a solvent, which one should think evaporates after deposition, should influence the dipole orientation and strength.
2) It is not completely clear to me If the values in figure 6 are extracted from the J-V curves in Figure S3. If this is the case, first of all, please explicitly mention it in the text, then let me say that paragraph 2.2 until the CdS and ZnO inclusion is to discard. It appears evident from figure S3 that the curves are so noisy that it is impossible to extract by fitting procedure both n and Rs that make sense. Furthermore, it appears that the devices were in some way under illumination, as the minimum of the curve is at positive voltages. Thus all the considerations must be reconsidered by re-measuring the devices in real dark conditions and with less noisy curves, otherwise, numbers extracted by fitting are difficult to be believed. Moreover, please when preparing figures consider that papers can be printed in b/w so add arrows or legends and use colours with high contrasts to allow also people printing papers in b/w to understand the figures (and this advice is valid also for figures 10-11).
3) Thicknesses. In the whole paper, the role of a dipole induced by PEI is discussed and recalled to justify various behaviour (WF modification in Silicon, electron transport, fill factor…) however it sounds in some cases a bit confusing, also because thicknesses are not well mentioned in the various discussions. For example, is the thickness the same for samples in figure 5 for resistance measurement, in which the solvent is varied? Have you measured? If yes, how?
What is the thickness of the layer in the Tauc plot in figure 3? Is the Eg of the material the same when produced with different solvents and concentrations?
On page 8, it is stated that different concentrations mean different thicknesses. According to the experimental section, the PEI is deposited by spin coating at 5000 rpm for 30 s. Would not be possible to modify the spinner parameters to obtain the same thickness for solutions with different concentrations? Moreover, would it be possible to know what is the thickness of the various samples? Or the difference in thickness with the concentration is just justified by your experience? Could not they be the same or very similar? When you talk about optimum concentration, does this optimum come from figure 6?
My concern also regards this binomial concentration-thickness: how can you discern if some characteristic depends on thickness or concentration if you have two variables? If you notice different thicknesses because of different concentrations of PEI films, why not work on an optimized thickness, equal for all of them so that a fair comparison can be done?
Anyway, on page 8 line 207 it is mentioned that if the thickness is thin, the dipole effect (of PEI or Al?) does not prevent the surface pinning from the metal (by the way, in this text part it is not clear of which structure is investigated unless one does not see figure S3). However, when discussing figure 4 the metal is mentioned to be responsible for the higher shift in the WF of silicon, thus of stronger accumulation due to the stronger dipole, and the barrier increases as the thickness decrease. Thus the two concepts appear in contrast, or maybe need to be better explained if we assume that the conclusion got from figure 6 is not wrong because of erroneous parameter estimation as I told in point 2).
I also noticed, on page 8 line 212 you then mention that a conduction mechanism can happen through thick PEI film, as the series resistance increases. How? Which could be the mechanism? Why it should happen just on thick films and not on thin?
On page 9 line 234 it is mentioned that when the buffer is employed, the thinner layer is better. Thus it is fundamental to determine where the dipole effect takes place. Is it due to the PEI bulk material? Is it at the interface with metal or buffer? Because if it is in the bulk of PEI, no matter what interface, solvent, or concentration is used, the relation of the effect must be linked with the thickness only, with the necessity of an estimation. If instead, it is due to the interface with metal, TCO or buffer then it is necessary to understand if it depends just on the presence of the interface or on the thickness of layers.
4) cells
In the abstract, I read the paper deals with the attempt of substituting the CdS buffer layer in chalcogenides-based cells by adopting ZnO and a polymer to produce higher voltages and increase the electron selectivity of these devices. Thus one could expect more emphasis on the ZnO part of the work. Instead, the part involving the ZnO is smaller than that dedicated to the polymer, but this is in accordance with the title.
I have a question about figure 7: does Table 1 refers to figure 7? or to figure S4? please better specify. Do you have an idea of the PEI thicknesses in figure 7? Are they all the same? Are they similar to one of Figure 6? Figure S4 refers to the fit of J-V curves of diodes with different PEI concentrations for CdS buffer. It is possible to see the J-V curves in the Supplementary info? Are the PEI thicknesses obtained with different concentrations similar to Figure 6? Have you made the same for ZnO?
Regarding the Jsc, I think it cannot depend on the PEI thickness in any way for two reasons: first, it has a large bandgap, so it should not absorb light and thus no trade-off is needed. If it absorbs something, then it is worth showing the Tauc plot, or the absorption coefficient, for each concentration, of course, once known the thickness. The second reason is that there is no trend in the Jsc values for the cells in the table.
Regarding the EQE I have a question: is the AZO layer the same for all the cells? if so, considering the different refractive indexes of ZnO and CdS, and different thicknesses of PEI, maybe the differences in Jsc are just due to different reflectances, so I would encourage authors to measure the reflectances and elaborate an IQE, which could give much more information about the device.
Finally, just a consideration on ZnO. There is a lacking of explanation on why it should work. results are presented with just a few rows of comments, and considering the abstract, I would expect more.
Round 2
Reviewer 2 Report
Dear Authors,
Thank you for your very hard work in improving the manuscript. I just have a few more requests, then the paper could be published, in my opinion. First, some minor issues:
- In the abstract acronyms should be introduced only if used in the abstract itself. Remove PEI at line 15.
- Line 62 maybe there is an overdue “n” after the parenthesis.
- Line 261you call Figure 4, but Figure 3 is called the first time at line 274. Figure S1 and Table S2 are not called in the manuscript (maybe they should be around line 217). Figure S3 is called at line 158… please check the order of figures and tables according to the order when they appear in the text.
- Thank you for table S6. Please add the dipole value in the table, or write the name of the solvents in the figure4 caption in whatever logical order (from lower to upper concentration, for instance). Fig 4b reports a diagram structure from another source. Please cite it appropriately.
- Thank you for the TLM plots, even if I meant the plots of Resistance vs spacing to see the linearity and the Rc value. Fig s5 do not add value to the paper. Please provide R vs d graphs or remove the actual figure.
- Thank you for the graph of thickness vs concentration. It seems to be not fully linear, especially when higher concentrations are involved, but it is ok in the context.
- Thank you for the clarification of the c-si substrate use. Please add in the manuscript the considerations about the amorphous silicon, why you call it amorphous even if there is methane and why you use it. The picture of the hexagonal cell is nice and helps in understanding, I would add it to the paper, even in SI if you prefer.
- Thank you for the explanation about why you don’t use methanol. If I got this question maybe also other readers will, thus please add it to the manuscript.
- Thank you for clarifying the thicknesses of Al in different samples. However, I have not found it in the text the information, thus please specify it in the manuscript.
- Thank you for the references about the WF of calcium and magnesium, please add to the manuscript to support your conclusions.
- Thank you for the explanation about dipoles. I see some of that info are now in the paper. Please add all the references you gave me in the paper, recalling in the text as “for further information see refs:….” or something similar. By the way, I see now in figure4 one of the band diagrams is reported. However please cite it properly and ask for reprint permission.
- Thank you for the improvement of the dark J-V, now they are far better.
- In figure 7, please specify the thickness is 0.5 nm.
Finally two important things:
1) Thicknesses.
Regarding thicknesses and what it is the "optimum", it seems that there is not any "optimum", as from reading the paper it is still not clear why an optimum of 0.01% of Fig 5 moves then to 0.001% in the conclusions, but the best cell is for 0.1% for the Voc and 0.05 for Jsc and FF. Thus I suggest to better explain, maybe inserting the following paragraph from your reply to my comments (or something similar):
“For instance, when aluminium is used as the electrode, it is optimal to use a relatively low concentration of b-PEI∼ ??????2??3/???/??= 0.01%. If one deposits this material on top a selective contact such as ??? or ??? before the aluminium, one enhances the contact with the electrode with a different material resulting in a different optimum value, in this case thinner ??????2??3/???/???/??= 0.001%.
In a sense, it is logical to think that, since this film affects the equilibrium state between the semiconductor and the electrode, if one changes any of them it requires a different optimum concentration. And this is what we see also for solar cells. Higher concentrations of b-PEI are necessary as we change the electrode from aluminium to Aluminium doped Zinc Oxide (AZO), thus reaching an optimum value of ???? ??2??3/???/???/??? = 0.1 %.”
Also, I have to admit that from this point of view, if this is the real conclusion, one could say that all the initial studies and work on finding some optimal for a structure which is not the cell is nonsense, and the only important thing is the study on the final cell structure…. Then please when saying “ Optimus thickness depends on the structure” please carefully discuss and justify this.
It is interesting also to note that the best cell of 0.1% is the best just because of Voc, while current and FF are best for 0.05. please comment on that, also citing this paragraph from your reply to my comments (or something similar): “From Transfer Matrix Method we calculate the reflectance of the stack AZO/PEI/ZnO over antimony selenide absorber. An increase in the PEI thickness generates an increase in the reflectivity. Thus, one should expect a reduction of the Jsc with the increase in the PEI layer thickness. Nevertheless, as we observe, the reduction in the Jsc is only shown where thicker layers, higher than 0.1%, are deposited. Also, we see from TMM that the reflectivity is lower when no PEI layer is present. Thus, the increase in the EQE when PEI layer is incorporated must come from electric properties rather than optical. The increase in selectivity introduced by the polymeric layer increases the proportion of electrons extracted per incident photon, even when the number of photons reaching the absorber is slightly lower due to the higher reflectivity”
2) Charge transport
Thank you for the information about the charge transport, in particular on the tunnelling probability. This indicates then that the tunnelling above 2 nm, 0.05% in figure 5, tunnelling is actually the main transport mechanism, but limits the transport itself. Please add on page 10 line 314 something like “predominant conduction mechanism, limiting the transport and increasing the resistivity in this structure.” … specify “this structure”, because then one looks at the cells and finds that higher concentration produces better currents and FF.
